# Impact of emergency department overcrowding on the occurrence of in-hospital cardiac arrest

Jin Hae Jun[1], Chae Ryoung Park[1], Incheol Park[1], Ji Hwan Lee[1], Yun Ho Roh[2], Min Joung Kim[1] *

**1** Department of Emergency Medicine, Yonsei University College of Medicine, Seoul, Republic of Korea,
**2** Biostatistics Collaboration Unit, Department of Biomedical Systems Informatics, Yonsei University College of Medicine, Seoul, Republic of Korea

☯ These authors contributed equally to this work.

\* boringzzz@yuhs.ac

**Data Availability Statement:** All relevant data are within the paper and its Supporting information files.

**Funding:** The author(s) received no specific funding for this work.

## Abstract

We aimed to determine whether emergency department (ED) overcrowding affects the occurrence of in-hospital cardiac arrest (IHCA) requiring resuscitation in the ED. This retrospective study was conducted in the ED of a single hospital. We applied the propensity score-matching method to adjust for differences in clinical characteristics in patients who visited the ED during overcrowded conditions. The indicators of overcrowding were: the total number of patients, number of patients undergoing treatment, and number of boarded patients awaiting hospital admission at the time of a patient's arrival. We defined the existence of ED overcrowding based on the 75%, 80%, and 90% thresholds of each indicator. We included 153,353 patients, and 160 cases of IHCA occurred, showing an incidence rate of 0.10%. The IHCA incidence rate increased during overcrowding, as indicated by the total number of patients and the number of boarded patients rising to 0.15% and 0.17%, respectively, at the 90% threshold (p = 0.0407 and 0.0203, respectively). The IHCA incidence rate did not increase during overcrowding based on the number of patients undergoing treatment. In the comparison conducted after propensity score matching, the incidence of IHCA was significantly higher in the overcrowding group than in the non-overcrowding group, indicated by 80% boarded patients (0.15% vs. 0.08%, p = 0.0222). The logistic regression results indicated that both the full-study and propensity score-matched cohorts showed a tendency for increased IHCA during overcrowding, as indicated by the total number of patients and number of boarded patients. The number of patients undergoing treatment did not affect the occurrence of IHCA. Although this needs to be confirmed in larger studies, we found in this study that ED overcrowding, particularly blocked access, tends to increase the incidence of IHCA requiring resuscitation in the ED. This suggests that to protect patient safety in ED overcrowding, it is essential for the entire hospital to make concerted efforts to maintain the flow of patients in the ED.

**Competing interests:** The authors have declared that no competing interests exist.

## Introduction

Emergency department (ED) overcrowding is a global problem that has persisted for decades [1, 2]. ED overcrowding not only leads to increased patient waiting times and delayed examinations but also causes psychological exhaustion among medical staff, hampers decision-making, and contributes to inappropriate decisions being made [3–5]. ED overcrowding poses a potentially lethal risk, particularly for critically ill patients. It can delay the initial evaluation of patients with acute stroke, increase mortality by postponing admission to the intensive care unit for mechanically-ventilated patients, and delay initial treatment for those with septic shock [6–8]. There is also a study that reported overnight stays in the ED for elderly patients increased mortality [9].

In-hospital cardiac arrest (IHCA) is an emergent situation requiring immediate treatment and continues to be a significant public health burden [10, 11]. The incidence of IHCA is approximately 1.6 out of 1,000 inpatients, with a survival rate of only 18.4% [11]. One study reported that 18% of IHCA cases can be prevented based on the fact that they occur owing to not treating patients according to established guidelines or failing to monitor patients [12]. ED overcrowding impedes the delivery of optimal care to patients, and studies have shown a correlation between ED overcrowding and the occurrence of IHCA [13, 14]. These studies have investigated the frequency of IHCA occurrences according to the state of ED overcrowding; however, they have not clarified whether overcrowding influences the incidence of IHCA. Considering that ED overcrowding increases ambulance diversion and the number of patients who leave without being seen, the clinical features of patients treated when the ED is overcrowded may show different characteristics than those in times when the ED is not overcrowded. However, to date, to our knowledge, no studies have investigated whether ED overcrowding, considering these patient differences, influences the occurrence of IHCA.

In this study, we aimed to determine whether ED overcrowding influences the incidence of IHCA. To prove this hypothesis, we employed the propensity score (PS)-matching method to adjust for clinical differences in patients according to the degree of ED overcrowding.

## Methods

### Study design and setting

This was a retrospective observational study carried out in the ED of a 2,000-bed tertiary hospital in a South Korean city, which receives 90,000 to 100,000 visits annually. This ED is divided into adult and pediatric sections. Our study was conducted in the adult ED, where patients aged ≥16 years are treated. The adult ED consists of a monitoring area with 16 beds, including three resuscitation beds, and a bed area with 29 beds, including three isolation beds, a chair area with 20 recliners, and a fast-track area. In Korea, the onset of the coronavirus disease 2019 (COVID-19) pandemic in February 2020 significantly impacted ED operations. EDs frequently faced temporary closures for quarantine purposes following the admission of infected patients. Additionally, external factors, such as the influx of patients due to closures of nearby hospitals, further exacerbated the challenges faced by these departments. Consequently, we established the study period from January 2018 to January 2020, prior to the COVID-19 outbreak.

IHCA was defined as sudden cardiac arrest requiring cardiopulmonary resuscitation during an ED stay. We excluded patients who experienced cardiac arrest before arriving at the ED, as well as visits for non-medical purposes, such as obtaining medical record copies. All data were extracted in an anonymized form from the hospital information system. This study was approved by the Institutional Review Board of the Yonsei University Health System Clinical

Trial Center (approval number: 4-2022-0670), and the requirement for patient informed consent was waived. The date of accessing the data for research purposes is July 14, 2022. The authors did not have access to information that could identify individual participants during or after data collection.

### Crowding indicators

To gauge ED overcrowding, we chose to use the total number of patients in the ED; number of patients receiving treatment, as indicators of ED input and throughput overload; and number of boarded patients, to reflect the ED exit block [15]. To acquire these indicators of ED overcrowding, we utilized three key time components of patient care flow: the time of ED arrival, of when a decision was made regarding patient disposition (either admission or discharge), and of ED departure for all patients treated during the study period. Using these time components, we reconstructed a dataset that counted the number of total patients in the ED, number of treated patients (from ED arrival to disposition decision), and number of boarded patients (from admission decision to transfer to a hospital ward) at 10-minute intervals (Fig 1).

It is likely that the adverse effects of ED overcrowding do not occur in proportion to the increase in patient numbers, but rather arise once a specific threshold level is surpassed. However, to our knowledge, previous studies have not explored this threshold. Consequently, we established nine indicators of overcrowding, defining thresholds at 75%, 80%, and 90% for the number of total patients, those in treatment, and boarded patients. If the value of overcrowding indicators surpassed these thresholds at the time of a patient's visit, the ED was considered overcrowded.

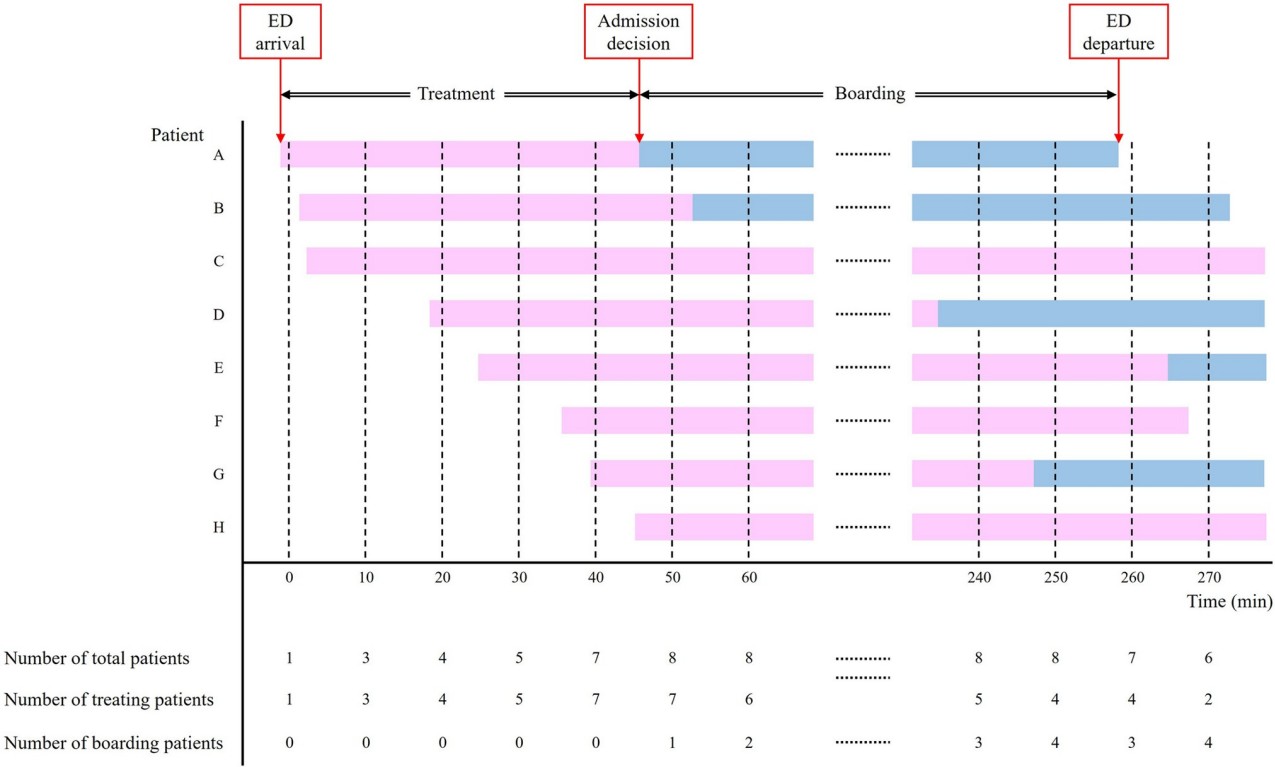

**Fig 1. Reconstructing indicators of emergency department overcrowding using time-based factors from individual patient data.** ED, emergency department.

## Study variables

The clinical characteristics of the patients were investigated through medical records. We investigated whether the patient arrived at the ED using emergency medical services and whether the patient was transferred after being treated at another hospital. The severity of the patients was assessed using the Korean Triage and Acuity Scale, a widely used system in Korea, where a triage nurse measures the severity on a five-level scale from resuscitation (1) to non-emergency (5) [16]. We assessed whether the patient's visit was related to non-medical issues, such as poisoning, trauma, or environmental factors. Complaint categories were determined according to the organ system associated with the primary symptom reported by the patient. Severe disease refers to a condition diagnosed with a critical diagnosis code, as designated by the Central Emergency Medical Center under the Ministry of Health and Welfare [17]. Additionally, we investigated the ED area where the patient was first assigned and treatment was initiated after triage or initial evaluation. We also investigated mental status and initial vital signs measured upon arrival at the ED, including systolic blood pressure, pulse rate, respiratory rate, oxygen saturation, and body temperature.

## Statistical analysis

In the full-study cohort, we created nine comparison groups (overcrowding and non-overcrowding) based on nine overcrowding indicators. We conducted 1:1 PS matching to mitigate confounding variables between the overcrowding and non-overcrowding groups. This process was repeated across all nine comparison groups. Caliper width was calculated as 0.2 of the standard deviation of the PS. The standardized mean differences (SMDs) were computed to evaluate differences between groups and to assess the balance of clinical characteristics of patients before and after PS matching [18]. A SMD of ≤0.1 was considered indicative of an appropriate balance among the covariates.

We compared patient characteristics between the overcrowding and non-overcrowding groups in the nine full-study and PS-matched cohorts. We presented the results as numbers and percentages and conducted comparisons using the chi-square test. To investigate the impact of ED overcrowding on IHCA incidence, we conducted multivariable logistic regression to adjust for confounding variables with a p-value <0.1 in the full-study cohort, and conditional logistic regression was performed in the PS-matched cohort. A p-value <0.05 was determined to be statistically significant. Data reconstruction to generate an overcrowding indicator was conducted with SAS (version 9.4; SAS Inc, Cary, NC, USA), and PS matching and statistical analysis were performed using the R package (version 4.0.3; http://www.R-project.org; R package 'matchit').

## Results

During the study period, an average of 201.86 patients visited the emergency department daily, with 47.50 admissions per day. The mean length of stay in the emergency department for all patients was 6.94 hours, and the average bed occupancy rate was 130.88% (Table 1).

There were 109,584 measurements of total, treated, and boarded patients generated at 10-minute intervals. The distribution of these numbers by day of the week and time of day is depicted in Fig 2. The number of total and treating patients was lowest at dawn and gradually increased to a peak in the afternoon, and there was a similar pattern for all days of the week. The number of boarded patients gradually increased during Monday and Tuesday; remained high on Wednesday, Thursday, and Friday; then began to decrease from Friday afternoon; and remained low over the weekend. The critical overcrowding points at 75%, 80%, and 90% of the total patient capacity were 67, 70, and 78 patients, respectively, with corresponding bed

**Table 1. Basic information on emergency department overcrowding.**

|  | Mean (95% CI) |
|---|---|
| **Daily visits, n** | 201.86 (199.66–204.05) |
| **Daily admissions, n** | 47.50 (46.87–48.13) |
| **Length of stay, hr** | 6.94 (6.89–6.99) |
| Admission patients | 15.22 (15.06–15.37) |
| Discharged patients | 4.40 (4.37–4.43) |
| **Treatment time, hr** | 4.50 (4.44–4.56) |
| **Boarding time, hr** | 9.11 (8.97–9.26) |
| **Bed occupancy rate, %** | 130.88 (130.70–131.06) |

occupancy rates of 171.8%, 179.5%, and 200.0%. Similarly, the numbers were 44, 46, and 52 for treated patients, and 24, 27, and 35 for boarded patients, respectively.

A total of 220,759 patients visited the ED during the study period. Of these, 66,155 patients aged <15 years, 928 patients with out-of-hospital cardiac arrest, and 323 patients who visited the hospital for issuance of medical records were excluded. Finally, 153,353 patients were included in the study (Fig 3). The overcrowding and non-overcrowding groups were divided based on nine overcrowding indicators, and from these full-study cohorts, nine PS-matched cohorts were derived through 1:1 PS matching.

Table 2 shows a comparison of the overcrowding and non-overcrowding groups based on 75% of the total number of patients. In the overcrowding group of the full-study cohort before PS matching, there were more patients aged ≥65 years, fewer patients using emergency medical services, and a higher proportion of patients transferred from other hospitals than those of the non-overcrowding group. The proportion of patients corresponding to Korean Triage and Acuity Scale scores of 1–3 was higher, more patients visited the ED owing to medical problems, and more patients were diagnosed with severe disease in the overcrowding group than those in the non-overcrowding group. In the overcrowding group, the proportion of patients

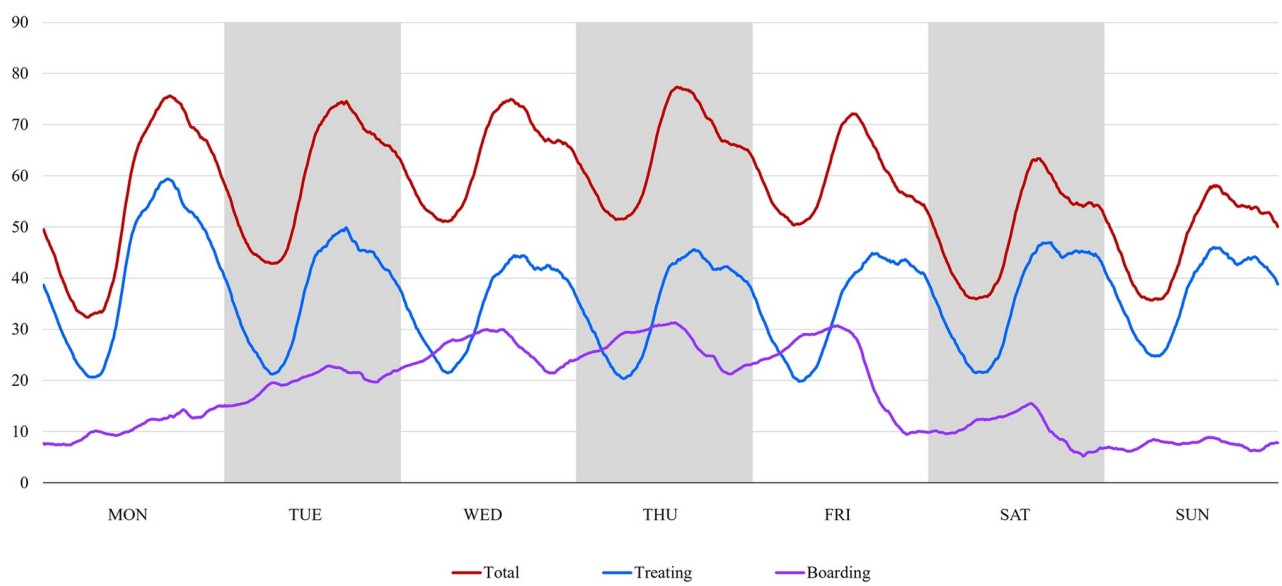

**Fig 2. Patterns of emergency department occupancy according to the day of the week and time of day: Analysis of total, treated, and boarded patient numbers.**

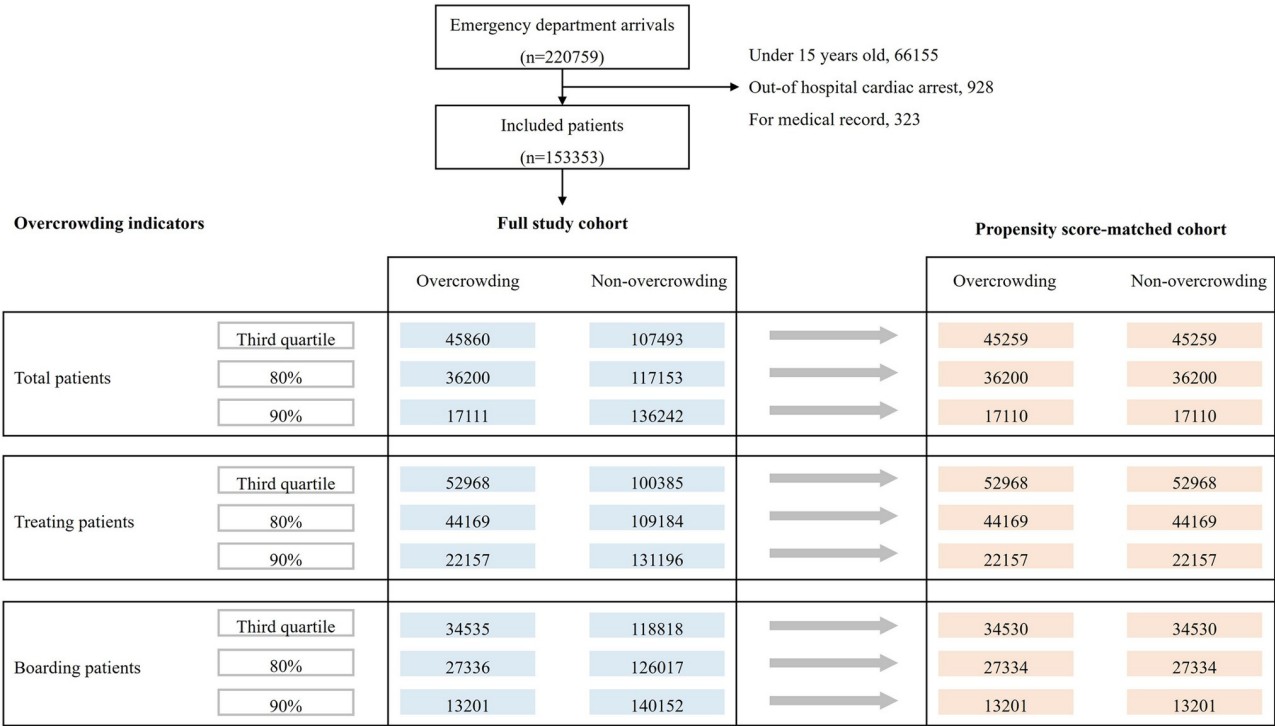

**Fig 3. Flowchart of study participants in the full-study and propensity score-matched cohorts.**

with systolic blood pressure <90 mmHg and patients with a fever of ≥38 ˚C was lower than in the non-overcrowding group. After PS matching, all parameters of basic characteristics were well-balanced between the two groups with an SMD <0.1. The results of patient characteristics of the full-study and PS-matched cohorts based on other overcrowding indicators are shown in S1–S8 Tables. All PS-matched cohorts were well-balanced between the two groups.

Out of the 153,353 patients, 160 experienced a sudden IHCA in the ED, resulting in an incidence rate of 0.10%. Table 3 compares the incidence rates of IHCA between the overcrowding and non-overcrowding groups in the full-study and PS-matched cohorts. In the overcrowding group, defined by the total number of patients, the incidence of IHCA was higher than in the non-overcrowding group at 0.12% at the 75% threshold, 0.14% at the 80% threshold, and 0.15% at the 90% threshold. When based on the number of boarded patients, the incidence of IHCA was higher than in the non-overcrowding group at 0.13% for the 75% threshold and 0.15% and 0.17% for the 80% and 90% thresholds, respectively. In the comparison conducted after propensity score matching, the incidence of IHCA was statistically significantly higher in the overcrowding group than in the non-overcrowding group, indicated by 80% boarded patients (0.15% vs. 0.08%, p = 0.0222).

Fig 4 shows the odds ratios for occurrence of IHCA in the full-study and PS-matched cohorts. Both cohorts showed a tendency of increased occurrence of IHCA in overcrowded conditions, with a large number of total patients and boarded patients. However, only the following two indicators in the full-study cohort showed statistically significant results: the odds ratio (95% CI) at the 80% threshold for total patients was 1.473 (1.034, 2.099), and for boarding patients at the 80% threshold, it was 1.475 (1.015, 2.144). In the PS-matched cohort, the results

**Table 2. Characteristics of patients in the full-study and propensity score-matched cohorts, stratified by emergency department overcrowding, based on the number of total occupying patients above the third quartile.**

| Variables | | Full-study cohort | | | | Propensity score-matched cohort | | | |
|---|---|---|---|---|---|---|---|---|---|
| | | Overcrowding (n = 45,860) | Non-overcrowding (n = 107,493) | SMD | p-value | Overcrowding (n = 45,259) | Non-overcrowding (n = 45,259) | SMD | p-value |
| Age (years) | ≤39 | 13,458 (29.35) | 37,076 (34.49) | -0.1130 | <0.0001 | 13,428 (29.67) | 14,165 (31.30) | -0.0358 | <0.0001 |
| | 40–64 | 16,662 (36.33) | 38,221 (35.56) | 0.0161 | | 16,334 (36.09) | 16,296 (36.01) | 0.0017 | |
| | 65–79 | 11,511 (25.10) | 23,610 (21.96) | 0.0723 | | 11,338 (25.05) | 10,859 (23.99) | 0.0244 | |
| | ≥80 | 4,229 (9.22) | 8,586 (7.99) | 0.0427 | | 4,159 (9.19) | 3,939 (8.70) | 0.0168 | |
| Male | | 21,542 (46.97) | 49,654 (46.19) | 0.0156 | 0.0050 | 21,273 (47.00) | 21,074 (46.56) | 0.0088 | 0.1850 |
| Emergency medical services | | 10,067 (21.95) | 27,462 (25.55) | -0.0869 | <0.0001 | 10,021 (22.14) | 9,864 (21.79) | 0.0084 | 0.2075 |
| Transfer in | | 8,183 (17.84) | 11,160 (10.38) | 0.1949 | <0.0001 | 7,700 (17.01) | 7,094 (15.67) | 0.0350 | <0.0001 |
| KTAS | 1 | 505 (1.10) | 1,136 (1.06) | 0.0043 | <0.0001 | 504 (1.11) | 505 (1.12) | -0.0002 | 0.0003 |
| | 2 | 3,935 (8.58) | 8,853 (8.24) | 0.0123 | | 3,883 (8.58) | 3,807 (8.41) | 0.0060 | |
| | 3 | 12,605 (27.49) | 26,096 (24.28) | 0.0719 | | 12,304 (27.19) | 11,736 (25.93) | 0.0281 | |
| | 4 | 23,395 (51.01) | 56,694 (52.74) | -0.0346 | | 23,155 (51.16) | 23,649 (52.25) | -0.0218 | |
| | 5 | 5,420 (11.82) | 14,714 (13.69) | -0.0579 | | 5,413 (11.96) | 5,562 (12.29) | -0.0102 | |
| Non-medical | | 6,936 (15.12) | 19,924 (18.54) | -0.0952 | <0.0001 | 6,935 (15.32) | 7,231 (15.98) | -0.0183 | 0.0068 |
| Chief complaints | Gastrointestinal | 9,115 (19.88) | 21,821 (20.30) | -0.0106 | <0.0001 | 8,947 (19.77) | 8,772 (19.38) | 0.0097 | 0.0008 |
| | General | 7,918 (17.27) | 17,096 (15.90) | 0.0360 | | 7,787 (17.21) | 7,891 (17.44) | -0.0061 | |
| | Neurological | 6,936 (15.12) | 15,498 (14.42) | 0.0197 | | 6,796 (15.02) | 6,706 (14.82) | 0.0056 | |
| | Cardiovascular | 4,937 (10.77) | 10,043 (9.34) | 0.0459 | | 4,841 (10.70) | 4,616 (10.20) | 0.0160 | |
| | Musculoskeletal | 4,181 (9.12) | 10,034 (9.33) | -0.0076 | | 4,176 (9.23) | 4,421 (9.77) | -0.0188 | |
| | Respiratory | 3,953 (8.62) | 7,365 (6.85) | 0.0630 | | 3,902 (8.62) | 3,708 (8.19) | 0.0153 | |
| | Skin | 2,627 (5.73) | 7,945 (7.39) | -0.0716 | | 2,627 (5.80) | 2,727 (6.03) | -0.0095 | |
| | ENT | 2,229 (4.86) | 7,100 (6.61) | -0.0811 | | 2,228 (4.92) | 2,286 (5.05) | -0.0060 | |
| | Others | 3,964 (8.64) | 10,591 (9.85) | -0.0430 | | 3,955 (8.74) | 4,132 (9.13) | -0.0139 | |
| Severe disease | | 5,628 (12.27) | 11,404 (10.61) | 0.0507 | <0.0001 | 5,561 (12.29) | 5,201 (11.49) | 0.0242 | 0.0002 |
| Area | Monitoring area | 3,917 (8.54) | 8,234 (7.66) | 0.0315 | <0.0001 | 3,915 (8.65) | 4,015 (8.87) | -0.0079 | 0.0388 |
| | Bed area | 7,040 (15.35) | 21,517 (20.02) | -0.1294 | | 7,040 (15.56) | 7,303 (16.14) | -0.0161 | |
| | Chair area | 2,337 (5.10) | 28,564 (26.57) | -0.9766 | | 2,337 (5.16) | 2,352 (5.20) | -0.0015 | |
| | Fast track | 32,566 (71.01) | 49,178 (45.75) | 0.5568 | | 31,967 (70.63) | 31,589 (69.80) | 0.0184 | |
| Mental status | Alert | 45,165 (98.48) | 105,558 (98.20) | 0.0233 | 0.0036 | 44,565 (98.47) | 44,624 (98.60) | -0.0107 | 0.5644 |
| | Drowsy | 486 (1.06) | 1,361 (1.27) | -0.0202 | | 485 (1.07) | 445 (0.98) | 0.0086 | |
| | Stupor | 133 (0.29) | 358 (0.33) | -0.0080 | | 133 (0.29) | 116 (0.26) | 0.0070 | |
| | Semi-comatose | 50 (0.11) | 140 (0.13) | -0.0064 | | 50 (0.11) | 48 (0.11) | 0.0013 | |
| | Coma | 26 (0.06) | 76 (0.07) | -0.0059 | | 26 (0.06) | 26 (0.06) | 0.0000 | |
| Systolic blood pressure (mmHG) | ≤89 | 3,418 (7.45) | 11,102 (10.33) | -0.1095 | <0.0001 | 3,409 (7.53) | 3,450 (7.62) | -0.0034 | 0.3578 |
| | 90–139 | 26,644 (58.10) | 60,055 (55.87) | 0.0452 | | 26,243 (57.98) | 26,030 (57.51) | 0.0095 | |
| | ≥140 | 15,798 (34.45) | 36,336 (33.80) | 0.0136 | | 15,607 (34.48) | 15,779 (34.86) | -0.0080 | |
| Pulse rate (bpm) | ≤59 | 1,366 (2.98) | 3,299 (3.07) | -0.0053 | 0.0025 | 1,359 (3.00) | 1,301 (2.88) | 0.0075 | 0.3467 |
| | 60–99 | 33,037 (72.04) | 78,217 (72.76) | -0.0162 | | 32,591 (72.01) | 32,751 (72.36) | -0.0079 | |
| | ≥100 | 11,457 (24.98) | 25,977 (24.17) | 0.0189 | | 11,309 (24.99) | 11,207 (24.76) | 0.0052 | |
| Respiratory rate (breaths/minute) | ≤11 | 169 (0.37) | 308 (0.29) | 0.0135 | 0.0262 | 167 (0.37) | 154 (0.34) | 0.0047 | 0.2455 |
| | 12–19 | 35,263 (76.89) | 82,580 (76.82) | 0.0016 | | 34,768 (76.82) | 34,968 (77.26) | -0.0105 | |
| | ≥20 | 10,428 (22.74) | 24,605 (22.89) | -0.0036 | | 10,324 (22.81) | 10,137 (22.39) | 0.0099 | |

(*Continued*)

**Table 2.** (Continued)

| Variables | | Full-study cohort | | | | Propensity score-matched cohort | | | |
|---|---|---|---|---|---|---|---|---|---|
| | | Overcrowding (n = 45,860) | Non-overcrowding (n = 107,493) | SMD | p-value | Overcrowding (n = 45,259) | Non-overcrowding (n = 45,259) | SMD | p-value |
| **Oxygen saturation (%)** | ≤89 | 495 (1.08) | 1,001 (0.93) | 0.0143 | <0.0001 | 495 (1.09) | 502 (1.11) | -0.0015 | 0.0050 |
| | 90–94 | 1,935 (4.22) | 3,879 (3.61) | 0.0304 | | 1,911 (4.22) | 1,719 (3.80) | 0.0211 | |
| | ≥95 | 43,430 (94.70) | 102,613 (95.46) | -0.0339 | | 42,853 (94.68) | 43,038 (95.09) | -0.0182 | |
| **Body temperature (°C)** | ≤35.9 | 1,319 (2.88) | 4,388 (4.08) | -0.0722 | <0.0001 | 1,316 (2.91) | 1,236 (2.73) | 0.0106 | 0.2540 |
| | 36.0–37.9 | 38,360 (83.65) | 85,809 (79.83) | 0.1032 | | 37,804 (83.53) | 37,914 (83.77) | -0.0066 | |
| | ≥38.0 | 6,181 (13.48) | 17,296 (16.09) | -0.0765 | | 6,139 (13.56) | 6,109 (13.50) | 0.0019 | |

SMD, standardized mean difference; KTAS, Korean Triage and Acuity Scale; ENT, ear, nose, and throat

were not statistically significant. When the number of treated patients was used as the overcrowding indicator, overcrowding was not a factor that increased IHCA incidence.

## Discussion

In this study, we analyzed the impact of ED overcrowding on IHCA occurrence by adjusting for patient characteristics during overcrowding using the PS-matching method. We established various overcrowding indicators, including the number of total patients, patients under treatment, and boarded patients, and confirmed that an excessive number of boarded patients had an impact on IHCA occurrence. However, a high number of patients under treatment did not have a significant impact on the occurrence of IHCA.

In previous studies, an association between ED overcrowding and the occurrence of IHCA has been reported [13, 14, 19]. Chang et al. [14] reported an association between the ED bed occupancy rate, which is the ratio of occupied beds to the total number of beds, and the occurrence of IHCA. Kim et al. [13] also found that the ED occupancy rate, which we also used as a measure of overcrowding, was associated with IHCA occurrence. Both studies, conducted using a time-based approach, suggest that IHCAs occur more frequently during periods of high overcrowding. However, considering that more patients are staying when EDs are overcrowded, it may be natural outcome that the frequency of IHCA is higher.

**Table 3. Comparison of in-hospital cardiac arrest incidence according to emergency department overcrowding in the full-study and propensity score-matched cohorts.**

| Overcrowding indicator | | Full study cohort | | | Propensity score-matched cohort | | |
|---|---|---|---|---|---|---|---|
| | | Overcrowding | Non-overcrowding | p-value | Overcrowding | Non-overcrowding | p-value |
| **Total patients** | 75% | 56/45,860 (0.12) | 104/107,493 (0.10) | 0.1590 | 56/45,259 (0.12) | 47/45,259 (0.10) | 0.3749 |
| | 80% | 50/36,200 (0.14) | 110/117,153 (0.09) | 0.0227 | 50/36,200 (0.14) | 36/36,200 (0.10) | 0.1309 |
| | 90% | 26/17,111 (0.15) | 134/136,242 (0.10) | 0.0407 | 26/17,110 (0.15) | 17/17,110 (0.10) | 0.1696 |
| **Treated patients** | 75% | 59/52,968 (0.11) | 101/100,385 (0.10) | 0.5343 | 59/52,968 (0.11) | 49/52,968 (0.09) | 0.3357 |
| | 80% | 44/44,169 (0.10) | 116/109,184 (0.11) | 0.7159 | 44/44,169 (0.10) | 36/44,169 (0.09) | 0.5830 |
| | 90% | 23/22,157 (0.10) | 137/131,196 (0.10) | 0.9789 | 23/22,157 (0.10) | 28/22,157 (0.13) | 0.4836 |
| **Boarded patients** | 75% | 44/34,535 (0.13) | 116/118,818 (0.10) | 0.1313 | 44/34,530 (0.13) | 40/34,530 (0.12) | 0.6623 |
| | 80% | 40/27,336 (0.15) | 120/126,017 (0.10) | 0.0177 | 40/27,334 (0.15) | 22/27,334 (0.08) | 0.0222 |
| | 90% | 22/13,201 (0.17) | 138/140,152 (0.10) | 0.0203 | 22/13,201 (0.17) | 13/13,201 (0.10) | 0.1279 |

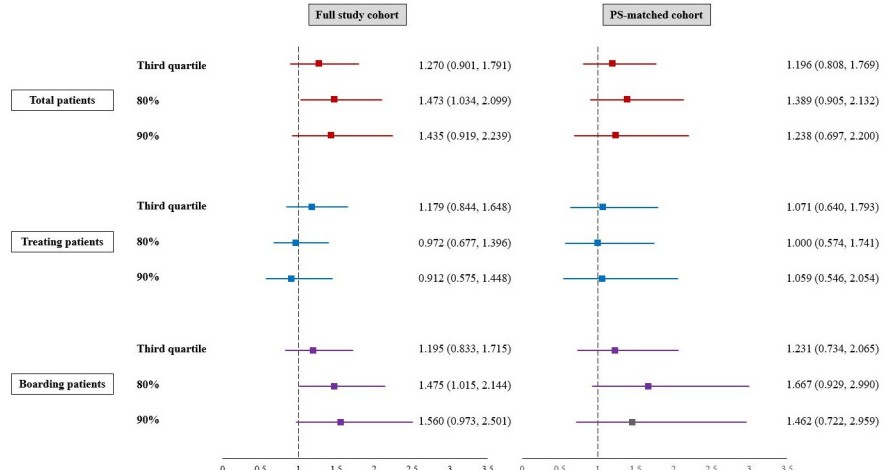

**Fig 4. Odds ratios for the incidence of in-hospital cardiac arrest according to emergency department overcrowding in the full-study and propensity score-matched cohorts.**

In order to determine the impact of ED overcrowding on the occurrence of IHCA, the larger number of patients during overcrowding than that in non-overcrowding must be corrected for, and further, differences in patient characteristics during overcrowding must also be considered. It is well known that ED overcrowding has an impact on factors related to patient input, such as increasing cases of ambulance diversion and patients leaving without being seen [20–22]. To the best of our knowledge, this study is the first to take into account the clinical characteristics of patients to investigate whether ED overcrowding influences the occurrence of IHCA.

In this study, we designated the number of total patients, number of patients in treatment, and number of boarded patients as indicators of ED overcrowding and confirmed that the number of total patients and number of boarded patients had an impact on the occurrence of IHCA. To prevent the occurrence of IHCA, it is essential for medical staff to promptly recognize the deteriorating condition of patients and administer emergency treatment swiftly [23, 24]. To ensure this, emergency beds and monitoring equipment must be accessible and made available to patients at risk of deterioration [25, 26]. Owing to the nature of the ED, controlling the influx of patients is challenging, often leading to situations where the demand for treatment exceeds available resources. Experienced emergency staff prepare to handle the arrival of seriously ill patients by maintaining some vacant emergency beds. However, in an access-block situation, emergency beds are already occupied by boarded patients who have been in the ED for an extended period, making it challenging for emergency staff to safely manage the ED with further diminished resources. In this context, it is evident that an access block increases the risk of IHCA in ED patients. An access block is a phenomenon caused by overcrowding throughout the hospital, a situation that cannot be resolved solely through the efforts of the ED [27, 28]. Therefore, levels of overcrowding that can cause adverse effects must be systematically managed at the hospital level [29–31].

Contrary to the number of boarded patients, the number of patients under treatment did not impact the incidence of IHCA. An access-block situation means that the flow of patients in the ED is obstructed, while overcrowding due to patients in treatment does not necessarily imply a blocked patient flow, despite a high number of patients. When there is overcrowding with the same number of patients, it becomes easier for medical staff to flexibly utilize

emergency resources when the patient flow is maintained. The results of previous studies, which indicate that access blocks result in more ambulance diversions than ED overcrowding itself, can be interpreted within the same context [32]. Another factor to consider when interpreting the phenomenon that the number of patients in treatment does not increase the risk of IHCA is that many EDs typically deploy additional medical personnel during peak times when there is a significant influx of patients [33, 34]. The augmentation of medical resources in anticipation of varying patient input patterns may be a factor that enhances the safety of treatment and helps prevent the occurrence of unexpected IHCA.

This study had several limitations. First, although we attempted to adjust for the clinical characteristics of patients using the PS-matching method, there remains the issue that unmeasured confounding factors could not be adjusted for owing to the limitations of the retrospective study design. Second, as this study was conducted at a single ED in a large urban university hospital, caution is required when interpreting the results. Since this ED frequently experiences overcrowding, the medical staff may have expertise in handling such conditions, which could have mitigated the adverse effects caused by overcrowding. Therefore, there is a concern that this study may have underestimated the impact of overcrowding on the occurrence of IHCA. Third, this study established overcrowding indicators based on the distribution of the total patient count, the number of patients undergoing tests, and the number of boarding patients to analyze the impact of various overcrowding factors in detail. Since these indicators are not commonly used overcrowding metrics, they may have limitations when comparing with other study results or applying them to other institutions' situations. Lastly, this study found that ED overcrowding tends to increase IHCA; however, the statistical significance appears to be somewhat limited. Due to the low incidence of HCA, insufficient sample sizes can result in wider confidence intervals for the odds ratio, making it difficult to achieve a statistically significant p-value. There is a need for large-scale studies that include a sufficient number of IHCA cases to obtain more reliable research results.

## Conclusion

In this study, we observed a trend towards increased occurrence of IHCA requiring resuscitation in the ED during ED overcrowding, although these findings should be confirmed in larger studies. Particularly, the finding that an access block negatively impacted the occurrence of IHCA highlights the critical importance of maintaining patient flow in the ED for patient safety. Therefore, to avoid the risk of IHCA caused by ED overcrowding, it is imperative that hospital-wide efforts focus on preventing exit block in the ED's patient flow.

## Supporting information

**S1 Table. Characteristics of patients in the full study cohort and the propensity score-matched cohort, stratified by emergency department overcrowding, based on the number of total occupying.**
(DOCX)

**S2 Table. Characteristics of patients in the full study cohort and the propensity score-matched cohort, stratified by emergency department overcrowding, based on the number of total occupying.**
(DOCX)

**S3 Table. Characteristics of patients in the full study cohort and the propensity score-matched cohort, stratified by emergency department overcrowding, based on the number**

of treating patients.
(DOCX)

**S4 Table. Characteristics of patients in the full study cohort and the propensity score-matched cohort, stratified by emergency department overcrowding, based on the number of treating patients.**
(DOCX)

**S5 Table. Characteristics of patients in the full study cohort and the propensity score-matched cohort, stratified by emergency department overcrowding, based on the number of treating patients.**
(DOCX)

**S6 Table. Characteristics of patients in the full study cohort and the propensity score-matched cohort, stratified by emergency department overcrowding, based on the number of boarding patients.**
(DOCX)

**S7 Table. Characteristics of patients in the full study cohort and the propensity score-matched cohort, stratified by emergency department overcrowding, based on the number of boarding patients.**
(DOCX)

**S8 Table. Characteristics of patients in the full study cohort and the propensity score-matched cohort, stratified by emergency department overcrowding, based on the number of boarding patients.**
(DOCX)

## Author Contributions

**Conceptualization:** Jin Hae Jun, Incheol Park, Min Joung Kim.

**Data curation:** Yun Ho Roh.

**Formal analysis:** Yun Ho Roh.

**Investigation:** Incheol Park.

**Methodology:** Chae Ryoung Park, Min Joung Kim.

**Supervision:** Incheol Park, Min Joung Kim.

**Validation:** Ji Hwan Lee, Min Joung Kim.

**Visualization:** Ji Hwan Lee.

**Writing – original draft:** Jin Hae Jun, Chae Ryoung Park.

**Writing – review & editing:** Min Joung Kim.

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
