## [Decision Letter · Decision Letter 0]

18 Sep 2024

PONE-D-24-27710Impact of emergency department overcrowding on the occurrence of in-hospital cardiac arrestPLOS ONE

Dear Dr. Kim,

Thank you for submitting your manuscript to PLOS ONE. After careful consideration, we feel that it has merit but does not fully meet PLOS ONE’s publication criteria as it currently stands. Therefore, we invite you to submit a revised version of the manuscript that addresses the points raised during the review process.

Thank you very much for having submitted this paper for consideration. The study is of potential interest as long as you solve the Reviewer's comments attached below.

We look forward to receiving your revised manuscript.

Kind regards,

Simone Savastano

Academic Editor

PLOS ONE

Journal Requirements:

Additional Editor Comments:

Thank you very much for having submitted this paper for consideration. The study is of potential interest as long as you solve the Reviewer's comments attached below.

Reviewers' comments:

Reviewer's Responses to Questions

**Comments to the Author**

1. Is the manuscript technically sound, and do the data support the conclusions?

Reviewer #1: No

Reviewer #2: Yes

Reviewer #3: Yes

2. Has the statistical analysis been performed appropriately and rigorously? 

Reviewer #1: No

Reviewer #2: Yes

Reviewer #3: Yes

3. Have the authors made all data underlying the findings in their manuscript fully available?

Reviewer #1: Yes

Reviewer #2: No

Reviewer #3: Yes

4. Is the manuscript presented in an intelligible fashion and written in standard English?

Reviewer #1: Yes

Reviewer #2: Yes

Reviewer #3: Yes

5. Review Comments to the Author

Reviewer #1: The authors used a retrospective electronic records review to test whether patients were more likely to suffer a cardiac arrest when overcrowding exists in the emergency department

The paper begins with the promise of testing whether an association exists between overcrowding in the ED and cardiac arrest.

They used a propensity score matching method to mitigate bias.

By the authors admission, using third quartile, 80, 90% thresholds is a novel way to approach this question. A problem with these thresholds is that it only applies to this hospital, and only to data collected during this time period. Imagine, for example, if in the future the number of patients in each category increases twofold. The thresholds would be the same, but they would describe an entirely different set of circumstances. I worry that the approach in this manuscript is an insurmountable barrier to interpretation and its applicability to other locations. Other studies that measure the percentage of beds occupied by boarders, or length of stay, can be compared to other sites and other time periods.

The bigger problem with this study is that the conclusions are not well supported by results. The authors overstate the results of what is essentially a null study. There are multiple comparisons made in Table 2/Figure 4. There are 9 comparisons in figure 4. Only one shows a confidence interval that does not cross 1, and that appears to be a fragile result, since the 95% CI is close to 1. If correction is made for multiple comparisons, the single significant result would be non-significant. One way that the authors could deal with this problem is to have a single pre-specified primary outcome variable. However, there is no evidence in this paper that the authors pre-specified the 80% percentile in boarding patients as the primary outcome variable.

Reviewer #2: I commend to the authors for their paper on the effect of ED overcrowding on IHCA.

This study covers a very important topic that - as authors state in the manuscript - is not only relevant for the community of emergency practitioners - but to the broader audience of clinicians and all stakeholders involved in patient safety at a hospital level.

With this retrospective study, the authors aimed to investigate the impact of ED overcrowding on IHCA.

What distinguishes this study from similar previous ones, is the use of propensity score matching to mitigate confounding variables between overcrowding and non-overcrowding groups.

I believe that this strategy is the most appropriate methodology for the research question, and represents one of the major strengths of the study.

Patients' characteristics used for the matching were also appropriately selected.

The background section covers adequately the pre-existing literature on the topic.

The methods section is detailed and clear.

Authors adequately report the limitations of the study and do not overestimate results.

Therefore, I think this study adds to current literature and should be considered for publication.

I only have a few minor suggestions:

- (Line 97 and 283) I would replace the word "obstruction/obstructed" with "blockage" or "exit block".

- In the background section I would add the recent French work on the effect of overnight ED boarding and mortality risk in the elderly population

(Roussel M, Teissandier D, Yordanov Y, et al. Overnight Stay in the Emergency Department and Mortality in Older Patients. JAMA Intern Med. 2023;183(12):1378–1385. doi:10.1001/jamainternmed.2023.5961)

- At line 253 I would add the following citation about the IFEM report on ED exit block

(Javidan AP, Hansen K, Higginson I, Jones P, Lang E; IFEM Task Force on Emergency Department Crowding, Access Block. The International Federation for Emergency Medicine report on emergency department crowding and access block: A brief summary. CJEM. 2021 Jan;23(1):26-28. doi: 10.1007/s43678-020-00065-9. Epub 2021 Jan 14. PMID: 33683618; PMCID: PMC7807403.)

- At line 206 the authors state that "Both cohorts showed a tendency of increased occurrence of IHCA in overcrowded conditions, with a large number of total patients and boarded patients".

It is true that there is a tendency of ORs towards increased occurrence of IHCA in overcrowded conditions, however the confidence intervals were very wide and often included the values of 1 or <1. Therefore, the evidence to reject the null hypotesis is weak.

Authors should detail this.

Reviewer #3: I would like to thank you for the chance to review the manuscript entitled "Impact of emergency department overcrowding on the occurrence of in-hospital cardiac arrest" by Kim et al., in which the authors explore the association between ED overcrowding and IHCA.

The manuscript is well written and scientific sound. The statistic is robust. I wouldn't be surprised, if the manuscript had already gone through peer-review. I have no comments to make and I endorse its publication.

6. PLOS authors have the option to publish the peer review history of their article (what does this mean?). If published, this will include your full peer review and any attached files.

Reviewer #1: No

Reviewer #2: **Yes: **Santi Di Pietro

Reviewer #3: No

---

## [Author Response · Author response to Decision Letter 0]

31 Oct 2024

Response to reviewers

On behalf of all the authors, I would like to thank you for helping us improve the quality of our manuscript. We have revised the manuscript in accordance with your recommendations. We would be grateful if you would consider the revised manuscript for publication.

Editor Comments:

Thank you very much for having submitted this paper for consideration. The study is of potential interest as long as you solve the Reviewer's comments attached below.

Reviewers' comments:

Reviewer #1: 

The authors used a retrospective electronic records review to test whether patients were more likely to suffer a cardiac arrest when overcrowding exists in the emergency department. The paper begins with the promise of testing whether an association exists between overcrowding in the ED and cardiac arrest. They used a propensity score matching method to mitigate bias.

By the authors admission, using third quartile, 80, 90% thresholds is a novel way to approach this question. A problem with these thresholds is that it only applies to this hospital, and only to data collected during this time period. Imagine, for example, if in the future the number of patients in each category increases twofold. The thresholds would be the same, but they would describe an entirely different set of circumstances. I worry that the approach in this manuscript is an insurmountable barrier to interpretation and its applicability to other locations. Other studies that measure the percentage of beds occupied by boarders, or length of stay, can be compared to other sites and other time periods.

--- We agree that your point is valid. Establishing our own unique overcrowding index based on patient distribution over a specific period may make it difficult to directly apply these figures to other institutions or studies. However, we hope you understand that we established indicators such as total patient count, number of patients under treatment, and number of boarding patients, as we wanted to conduct a detailed analysis of the impact of each overcrowding factor—input, throughput, and output. Explaining our ED's overcrowding situation in a more generalized way could help alleviate this issue, so we have analyzed and presented the overcrowding information into the results. We have also noted in the limitations section that our method of defining overcrowding restricts the comparability and applicability of our results.

(in the Result section)

During the study period, an average of 201.86 patients visited the emergency department daily, with 47.50 admissions per day. The mean length of stay in the emergency department for all patients was 6.94 hours, and the average bed occupancy rate was 130.88% (Table 1).

Table 1. Basic information on emergency department overcrowding

 Mean (95% CI)

Daily visits, n 201.86 (199.66-204.05)

Daily admissions, n 47.50 (46.87-48.13)

Length of stay, hr 6.94 (6.89-6.99)

Admission patients 15.22 (15.06-15.37)

Discharged patients 4.40 (4.37-4.43)

Treatment time, hr 4.50 (4.44-4.56)

Boarding time, hr 9.11 (8.97-9.26)

Bed occupancy rate, % 130.88 (130.70-131.06)

The critical overcrowding points at 75%, 80%, and 90% of the total patient capacity were 67, 70, and 78 patients, respectively, with corresponding bed occupancy rates of 171.8%, 179.5%, and 200.0%.

(in the Limitation section)

Third, this study established overcrowding indicators based on the distribution of the total patient count, the number of patients undergoing tests, and the number of boarding patients to analyze the impact of various overcrowding factors in detail. Since these indicators are not commonly used overcrowding metrics, they may have limitations when comparing with other study results or applying them to other institutions' situations.

The bigger problem with this study is that the conclusions are not well supported by results. The authors overstate the results of what is essentially a null study. There are multiple comparisons made in Table 2/Figure 4. There are 9 comparisons in figure 4. Only one shows a confidence interval that does not cross 1, and that appears to be a fragile result, since the 95% CI is close to 1. If correction is made for multiple comparisons, the single significant result would be non-significant. One way that the authors could deal with this problem is to have a single pre-specified primary outcome variable. However, there is no evidence in this paper that the authors pre-specified the 80% percentile in boarding patients as the primary outcome variable.

--- We acknowledge that we did not provide sufficient explanation regarding the aspects of weak statistical significance in the study results. This issue was also mentioned by another reviewer, and we consider it a very important point of critique. Overcrowding appears to have a tendency to impact IHCA incidence, though statistical significance is low. This outcome may be influenced by the low incidence of IHCA, indicating the need for larger-scale studies to achieve more reliable statistical analysis. We have revised the results and discussion sections as follows to ensure that our findings are not overstated and that the limitations are clearly recognized.

(in the Result section)

However, only the following two indicators in the full-study cohort showed statistically significant results: the odds ratio (95% confidence interval) at the 80% threshold for total patients was 1.473 (1.034, 2.099), and for boarding patients at the 80% threshold, it was 1.475 (1.015, 2.144). In the PS-matched cohort, the results were not statistically significant.

(in the Limitation section)

Lastly, this study found that ED overcrowding tends to increase IHCA; however, the statistical significance appears to be somewhat limited. Due to the low incidence of HCA, insufficient sample sizes can result in wider confidence intervals for the odds ratio, making it difficult to achieve a statistically significant p-value. There is a need for large-scale studies that include a sufficient number of IHCA cases to obtain more reliable research results.

(in the Conclusion section)

In this study, we observed that ED overcrowding tends to increase the occurrence of IHCA requiring resuscitation in the ED.

(in the Abstract)

ED overcrowding, especially access blockage, tends to increase the occurrence of IHCA requiring resuscitation in the ED.

Reviewer #2: 

I commend to the authors for their paper on the effect of ED overcrowding on IHCA. This study covers a very important topic that - as authors state in the manuscript - is not only relevant for the community of emergency practitioners - but to the broader audience of clinicians and all stakeholders involved in patient safety at a hospital level.

With this retrospective study, the authors aimed to investigate the impact of ED overcrowding on IHCA. What distinguishes this study from similar previous ones, is the use of propensity score matching to mitigate confounding variables between overcrowding and non-overcrowding groups. I believe that this strategy is the most appropriate methodology for the research question, and represents one of the major strengths of the study. Patients' characteristics used for the matching were also appropriately selected. 

The background section covers adequately the pre-existing literature on the topic. The methods section is detailed and clear. Authors adequately report the limitations of the study and do not overestimate results. Therefore, I think this study adds to current literature and should be considered for publication.

I only have a few minor suggestions:

- (Line 97 and 283) I would replace the word "obstruction/obstructed" with "blockage" or "exit block".

--- Thank you for suggesting a more appropriate word. We have revised it as follows.

To gauge ED overcrowding, we chose to use the total number of patients in the ED; number of patients receiving treatment, as indicators of ED input and throughput overload; and number of boarded patients, to reflect the ED exit block.

Therefore, to avoid the risk of IHCA caused by ED overcrowding, it is imperative that hospital-wide efforts focus on preventing exit block in the ED's patient flow.

- In the background section I would add the recent French work on the effect of overnight ED boarding and mortality risk in the elderly population.

(Roussel M, Teissandier D, Yordanov Y, et al. Overnight Stay in the Emergency Department and Mortality in Older Patients. JAMA Intern Med. 2023;183(12):1378–1385. doi:10.1001/jamainternmed.2023.5961)

--- Thank you for suggesting a recent and relevant reference. We have added it as follows.

There is also a study that reported overnight stays in the ED for elderly patients increased mortality [9].

- At line 253 I would add the following citation about the IFEM report on ED exit block

(Javidan AP, Hansen K, Higginson I, Jones P, Lang E; IFEM Task Force on Emergency Department Crowding, Access Block. The International Federation for Emergency Medicine report on emergency department crowding and access block: A brief summary. CJEM. 2021 Jan;23(1):26-28. doi: 10.1007/s43678-020-00065-9. Epub 2021 Jan 14. PMID: 33683618; PMCID: PMC7807403.)

--- We have added the journal you suggested. Thank you for your thoughtful advice.

- At line 206 the authors state that "Both cohorts showed a tendency of increased occurrence of IHCA in overcrowded conditions, with a large number of total patients and boarded patients".

It is true that there is a tendency of ORs towards increased occurrence of IHCA in overcrowded conditions, however the confidence intervals were very wide and often included the values of 1 or <1. Therefore, the evidence to reject the null hypothesis is weak. Authors should detail this.

--- In agreement with your perspective, we have added the following details in the results and discussion sections to ensure that the statistical findings are not interpreted in an exaggerated manner.

(in the Result section)

However, only the following two indicators in the full-study cohort showed statistically significant results: the odds ratio (95% confidence interval) at the 80% threshold for total patients was 1.473 (1.034, 2.099), and for boarding patients at the 80% threshold, it was 1.475 (1.015, 2.144). In the PS-matched cohort, the results were not statistically significant.

(in the Limitation section)

Lastly, this study found that ED overcrowding tends to increase IHCA; however, the statistical significance appears to be somewhat limited. Due to the low incidence of HCA, insufficient sample sizes can result in wider confidence intervals for the odds ratio, making it difficult to achieve a statistically significant p-value. There is a need for large-scale studies that include a sufficient number of IHCA cases to obtain more reliable research results.

(in the Conclusion section)

In this study, we observed that ED overcrowding tends to increase the occurrence of IHCA requiring resuscitation in the ED.

(in the Abstract)

ED overcrowding, especially access blockage, tends to increase the occurrence of IHCA requiring resuscitation in the ED.

Reviewer #3: 

I would like to thank you for the chance to review the manuscript entitled "Impact of emergency department overcrowding on the occurrence of in-hospital cardiac arrest" by Kim et al., in which the authors explore the association between ED overcrowding and IHCA. 

The manuscript is well written and scientific sound. The statistic is robust. I wouldn't be surprised, if the manuscript had already gone through peer-review. I have no comments to make and I endorse its publication.

--- We sincerely appreciate your positive feedback and the time and effort you dedicated to reviewing our manuscript. Your recognition of its clarity and the robustness of the statistical analysis is greatly valued. Thank you for your encouragement regarding its publication.

---

## [Decision Letter · Decision Letter 1]

29 Dec 2024

Impact of emergency department overcrowding on the occurrence of in-hospital cardiac arrest

PONE-D-24-27710R1

Dear Dr. Kim,

We’re pleased to inform you that your manuscript has been judged scientifically suitable for publication and will be formally accepted for publication once it meets all outstanding technical requirements.

Kind regards,

Simone Savastano

Academic Editor

PLOS ONE

Additional Editor Comments (optional):

Dear Authors, thank you much for having submitted this paper and addressed the Reviewers's comments. Now the quality and clarity of your work has increased and it is now suitable for publication in PLOS One.

Reviewers' comments:

Reviewer's Responses to Questions

**Comments to the Author**

1. If the authors have adequately addressed your comments raised in a previous round of review and you feel that this manuscript is now acceptable for publication, you may indicate that here to bypass the “Comments to the Author” section, enter your conflict of interest statement in the “Confidential to Editor” section, and submit your "Accept" recommendation.

Reviewer #2: (No Response)

Reviewer #3: All comments have been addressed

2. Is the manuscript technically sound, and do the data support the conclusions?

Reviewer #2: Yes

Reviewer #3: Yes

3. Has the statistical analysis been performed appropriately and rigorously? 

Reviewer #2: Yes

Reviewer #3: Yes

4. Have the authors made all data underlying the findings in their manuscript fully available?

Reviewer #2: No

Reviewer #3: Yes

5. Is the manuscript presented in an intelligible fashion and written in standard English?

Reviewer #2: Yes

Reviewer #3: Yes

6. Review Comments to the Author

Reviewer #2: Thank you for addressing all comments. I believe that the sentence below is still an overstatement based on the statistical weakness of the results previously discussed. Additionaly, the way the sentence is built implies causality instead of describing an association.

(in the Conclusion section)

In this study, we observed that ED overcrowding tends to increase the occurrence of IHCA requiring resuscitation in the ED.

I would suggest amending it as follows:

"In this study, we observed a trend towards increased occurrence of IHCA requiring resuscitation in the ED during ED overcrowding, although these findings should be confirmed in larger studies".

The same applies to the following sentence in the Abstract, in which there is a strong imbalance towards causality:

(in the Abstract)

ED overcrowding, especially access blockage, tends to increase the occurrence of IHCA requiring resuscitation in the ED.

I believe that the paper will be suitable for publication following amendments of the above sentences.

Once again congratulations for this important research.

Reviewer #3: I have no further comments to make. I believe that the manuscript is sound and the topic of interest.

7. PLOS authors have the option to publish the peer review history of their article (what does this mean?). If published, this will include your full peer review and any attached files.

Reviewer #2: **Yes: **Santi Di Pietro

Reviewer #3: No

---

## [Editor Report · Acceptance letter]

8 Jan 2025

PONE-D-24-27710R1 

PLOS ONE

Dear Dr. Kim, 

I'm pleased to inform you that your manuscript has been deemed suitable for publication in PLOS ONE. Congratulations! Your manuscript is now being handed over to our production team.

Kind regards, 

on behalf of

Dr. Simone Savastano 

Academic Editor

PLOS ONE